# Myostatin Modulation in Spinal Muscular Atrophy: A Systematic Review of Preclinical and Clinical Evidence

**DOI:** 10.3390/ijms26125858

**Published:** 2025-06-18

**Authors:** Martina Gnazzo, Giulia Pisanò, Valentina Baldini, Giovanna Giacomelli, Silvia Scullin, Benedetta Piccolo, Emanuela Claudia Turco, Susanna Esposito, Maria Carmela Pera

**Affiliations:** 1Department of Biomedical, Metabolic and Neural Sciences, University of Modena and Reggio Emilia, 41121 Modena, Italy; martignazzo@hotmail.it (M.G.); giuliapisan@gmail.com (G.P.); 2Department of Biomedical and Neuromotor Sciences, University of Bologna, 40127 Bologna, Italy; 3Pediatric Clinic, University Hospital, Department of Medicine and Surgery, University of Parma, 43126 Parma, Italy; giovanna.giacomelli@unipr.it (G.G.); silvia.scullin@unipr.it (S.S.); susannamariaroberta.esposito@unipr.it (S.E.); 4Department of Medicine and Surgery, Child Neuropsychiatry, University of Parma, 43126 Parma, Italy; bpiccolo@ao.pr.it (B.P.); eturco@ao.pr.it (E.C.T.)

**Keywords:** spinal muscular atrophy, myostatin inhibition, SMN-targeted therapy, muscle atrophy, apitegromab, motor function restoration, adjunctive treatment strategy

## Abstract

Spinal Muscular Atrophy (SMA) is a genetic disorder characterized by the progressive loss of motor neurons and consequent muscle atrophy. Although SMN-targeted therapies have significantly improved survival and motor outcomes, residual muscle weakness remains a major clinical challenge, particularly in patients treated later in the disease course. Myostatin, a potent negative regulator of skeletal muscle mass, has emerged as a promising therapeutic target to address this gap. This review summarizes the preclinical and clinical evidence supporting the modulation of the myostatin pathway in SMA. Preclinical studies have demonstrated that inhibiting myostatin, especially when combined with SMN-enhancing agents, can increase muscle mass, improve motor function, and enhance neuromuscular connectivity in SMA mouse models. These findings provide a strong rationale for translating myostatin inhibition into clinical practice as an adjunctive strategy. Early clinical trials investigating myostatin inhibitors have shown favorable safety profiles and preliminary signs of target engagement. However, large-scale trials have yet to demonstrate widespread, robust efficacy across diverse patient populations. Despite this, myostatin pathway inhibition remains a compelling approach, particularly when integrated into broader treatment paradigms aimed at enhancing motor unit stability and function in individuals with SMA. Further clinical research is essential to validate efficacy, determine optimal timing, and define the patient subgroups most likely to benefit from myostatin-targeted therapies.

## 1. Introduction

Myostatin (MSTN), also known as growth and differentiation factor 8 (GDF-8), is a secreted protein belonging to the transforming growth factor β (TGF-β) superfamily and serves as a powerful negative regulator of skeletal muscle growth. First discovered in 1997 by McPherron and Lee, MSTN is primarily expressed in skeletal muscle both during embryonic development and adult life, though it is also produced in small amounts by adipose tissue and cardiac muscle [1,2,3,4]. Its biological function has been conserved across mammalian species, and natural mutations in the MSTN gene are associated with a hypermuscular phenotype in animals such as cattle, dogs, sheep, and even humans. In contrast, overexpression of MSTN is implicated in muscle atrophy, highlighting its central role in maintaining muscle mass homeostasis [3,5,6,7,8]. Myostatin is synthesized as an inactive precursor protein (proMSTN), which undergoes proteolytic cleavage by Furin-like and BMP-1/tolloid (TLD)-like metalloproteinases to generate a mature, biologically active form [1,9,10,11]. Once activated, MSTN binds to activin type II receptors (ActRIIA/B), preferentially ActRIIB, initiating intracellular signaling cascades that activate SMAD2/3 transcription factors and the MAPK (JNK, ERK, and p38) pathways, ultimately inhibiting myogenesis and promoting the expression of muscle-specific atrophy genes such as Atrogin1 and MuRF1 [12,13,14,15,16,17,18].

Given its pivotal role in negatively regulating muscle development, myostatin has become a compelling target of research in the context of Spinal Muscular Atrophy (SMA)—a severe, genetically inherited neuromuscular disorder characterized by progressive degeneration of α-motor neurons in the anterior horn of the spinal cord, leading to denervation-induced skeletal muscle atrophy, weakness, and, in severe cases, paralysis [19,20]. The disease, most commonly caused by homozygous deletions or mutations in the SMN1 gene located on chromosome 5q13, leads to insufficient levels of the survival motor neuron (SMN) protein. Although a nearby gene, SMN2, can partially compensate, it primarily produces a truncated, non-functional protein due to alternative splicing that excludes exon 7. The number of SMN2 gene copies serves as a critical disease modifier, with higher copy numbers generally correlating with milder phenotypes. Despite the advent of SMN-enhancing therapies such as the antisense oligonucleotide Nusinersen, gene therapy Onasemnogene Abeparvovec, and splicing modifier Risdiplam, these treatments are most effective when initiated presymptomatically, often yielding incomplete clinical responses in patients diagnosed later in life or with fewer SMN2 copies.

This therapeutic gap has prompted interest in SMN-independent mechanisms of disease progression and the exploration of adjunctive treatments that target muscle tissue directly. In this context, MSTN emerges as both a biomarker and a potential therapeutic target to mitigate muscle wasting in symptomatic SMA patients. Elevated MSTN activity may exacerbate muscle atrophy in the denervated muscles of SMA patients, while its inhibition has shown promise in animal models as a strategy to restore or preserve skeletal muscle mass.

Preliminary results from a recent Phase 2 clinical trial investigating Apitegromab, a selective anti-promyostatin monoclonal antibody, in patients with SMA types 2 and 3 (NCT03921528), have demonstrated encouraging outcomes. Improvements in motor function observed at 12 months were sustained through 36 months of treatment [21,22]. The safety and efficacy of Apitegromab and other myostatin pathway inhibitors are now being evaluated in larger Phase 3 studies. Additional investigational therapies include GYM329, an anti-latent myostatin sweeping antibody developed by Roche (NCT05115110), and Taldefgrobep alfa, a bivalent, humanized anti-myostatin adnectin fused with a human IgG1 Fc domain, under development by Biohaven (NCT05337553) [23,24]. These novel agents represent a promising direction in the management of SMA, particularly for patients with limited response to SMN-directed therapies.

Consequently, this review aims to explore the regulatory functions of myostatin in muscle tissue, its pathological involvement in SMA-related muscle degeneration, and its utility as both a prognostic biomarker and a therapeutic target. In doing so, we hope to highlight emerging strategies that could improve motor function and quality of life in patients who fall outside the optimal treatment window for SMN-enhancing therapies.

## 2. Relevant Sections

### 2.1. Method

This systematic literature review was conducted and reported in accordance with the Preferred Reporting Items for Systematic Reviews and Meta-Analyses (PRISMA) guidelines [25]. The study protocol was registered in advance with PROSPERO (CRD420251052920).

#### 2.1.1. Search Strategy

A systematic search was conducted on PubMed using the following search criteria “spinal muscular atrophy” AND (myostatin OR “growth differentiation factor 8”).

#### 2.1.2. Selection of the Studies

During the screening phase, all relevant original research articles were identified based on their titles and abstracts. At this stage, articles without pertinent information on the subject, including those not in English, reviews, conference abstracts, editorials, and viewpoints, were excluded. Full texts of the articles derived from the screening phase were reviewed to determine whether they met the selection criteria. These full texts were also searched manually to identify additional studies. Articles that were incomplete and those that did not meet the objectives were excluded.

#### 2.1.3. Data Extraction

We extracted data and we checked them based on the characteristics of the studies. For Table 1 the following were extracted: Title, Author(s), Year, Country, Study Design, Population Type, SMA Type. The absence of data in some columns indicates that the specific information was not reported in the original source. For the Table 2 the data extracted were: Author(s), Year, SMA Type, Treatment Route and Dosage, Key Molecular Findings, Functional Outcomes, and Histological Outcomes.

#### 2.1.4. Assessment of the Quality of Studies and Risk of Bias from the Review

The STROBE Checklist for cohort, case-control, and cross-sectional studies (combined) was used to rate the quality of studies included in this review [37]. Based on these criteria, the studies included in the review were judged to be of good, moderate, or poor quality. The Risk of Bias in Systematic Reviews tool was used to ascertain the risk of bias arising from the quality of included studies or the methods of this review [38].

To reduce the selection bias arising from included studies and the bias in rating the quality of studies, these procedures were initially carried out independently by two authors. Any discrepancies were resolved by joint consensus following the independent evaluations.

### 2.2. Results

#### 2.2.1. Study Selection

Search results are summarized in the Prisma Flow Chart in Figure 1 Initially, one hundred and ten studies were identified through the database search strategy, and one study was identified through manual search. Ninety-seven studies remained after removing duplicates. After title and abstract screening, twenty studies remained. Finally, we included thirteen studies in the systematic review.

#### 2.2.2. Preclinical and Clinical Evidence

This review synthesizes findings from both preclinical and clinical studies investigating myostatin modulation in Spinal Muscular Atrophy (Figure 2). Preclinical investigations, primarily employing the C/C or A7 mouse models of SMA, consistently demonstrate improved muscle phenotypes and enhanced motor function through myostatin pathway inhibition [29,34]. Specifically, studies modulating myostatin through sActRIIB or dnMstn resulted in increased muscle mass (as measured by increased fiber cross-sectional area, CSA) in affected muscles such as the tibialis anterior (TA) and extensor digitorum longus (EDL). While muscle-specific improvements were clear, a significant recovery within functional metrics has not been observed in all such studies, with evidence of varying degrees of success that may indicate issues with what happens due to those processes. This review, however, also synthesized data from those testing SMN as a direct and improved approach [28]. Data on key molecular findings indicate that the level of SMAD3 was elevated, which may have been attributed to some other benefits when the protein is enhanced [29]. By that, a greater understanding is shown overall. Clinical studies with Apitegromab point to what happened during use [21,31]. In that context, this may provide a direction that enhances clinical utility and better management to provide treatment better overall. There will also be lower levels related to HINE compared to RHLM. There remains an important question on exactly how each agent may connect towards results.

## 3. Discussion

This review synthesizes preclinical and clinical evidence examining the therapeutic potential of myostatin inhibition in SMA, with particular attention to its mechanistic action, functional outcomes, and translational relevance. The convergence of findings across animal models and human trials suggests that targeting the myostatin pathway may represent a biologically sound and clinically promising adjunct to SMN-enhancing therapies, particularly for addressing muscle atrophy and persistent motor dysfunction.

This review prioritized the myostatin pathway over other anabolic mechanisms, such as IGF-1/AKT signaling, due to its central role in negatively regulating skeletal muscle mass and its disease-specific relevance in SMA. Unlike IGF-1/AKT, which promotes general anabolic effects, myostatin inhibition directly counteracts the atrophic signaling cascade activated by denervation. Additionally, preclinical SMA models have shown that myostatin blockade improves neuromuscular architecture and muscle fiber size independently of SMN restoration, reinforcing its therapeutic plausibility in this context.

Across the reviewed studies, several consistent themes emerge. First, preclinical models—including SMNΔ7 mice and pharmacologically induced A7 models—demonstrate that myostatin blockade increases muscle mass, improves neuromuscular junction (NMJ) integrity, and enhances motor performance [21]. These effects occur even in the absence of elevated SMN protein levels, indicating that myostatin inhibition acts through an SMN-independent mechanism, likely by reversing disuse muscle atrophy and stimulating anabolic signaling pathways. Therefore, the dual targeting of SMN and myostatin offers a complementary therapeutic strategy: while SMN-enhancing agents aim to rescue motor neuron integrity and synaptic transmission, myostatin inhibitors directly address denervation-induced muscle atrophy by promoting muscle fiber growth and neuromuscular junction stability. This combined approach is particularly promising for patients who initiate treatment beyond the early neurodevelopmental window, when irreversible motor neuron loss has already occurred. This distinction is especially relevant given the limitations of SMN-directed treatments, which—while effective in halting motor neuron degeneration—do not fully restore motor capacity in most patients, especially when treatment is initiated later in the disease course.

Secondly, clinical trials have begun to translate these preclinical gains into patient benefits. Studies such as those conducted by Barrett et al. provide early clinical evidence supporting the functional relevance of myostatin inhibition in SMA types 2 and 3 [26]. Apitegromab, a monoclonal antibody that selectively binds and inhibits the precursor forms of myostatin, has shown a favorable safety profile in both healthy individuals and SMA patients. In the TOPAZ Phase 2 trial, Apitegromab exhibited dose-dependent pharmacodynamic effects and clinically meaningful improvements in motor function, with results not only observed at 12 months but also sustained through 36 months of follow-up, particularly in non-ambulatory patients and those with limited response to SMN-enhancing agents [26]. Notably, the selection of SMA type 2 and 3 patients was based on clinical phenotype, age, and motor function level, rather than SMN2 copy number, to allow the evaluation of myostatin inhibition in a representative population already receiving SMN-targeted therapies. This strategy aligns with the trial’s goal of improving functional outcomes in individuals with established disease, regardless of their genetic SMN2 background.

Importantly, this review also highlights variability in treatment response. For example, Servais et al.’s Phase 3 investigation of Taldefgrobep alfa—another myostatin-targeting compound—did not reproduce the positive outcomes seen with Apitegromab [23]. This divergence may be attributed to differences in molecular targeting, drug half-life, dosing regimens, or study populations. These contrasting findings underscore the importance of refining patient selection and understanding pharmacological distinctions across different myostatin inhibitors. In this context, anti-myostatin antibodies such as Apitegromab have been selected for their upstream mechanism of action, which allows for the selective inhibition of myostatin activation by targeting its precursor form, without interfering with mature growth factors or other TGF-β superfamily ligands. This contrasts with latent myostatin-targeting agents, such as GYM329, which may have broader effects and less specificity. The selective binding profile of Apitegromab is associated with a favorable safety and tolerability profile, particularly important for chronic treatment in pediatric SMA populations.

Despite the significant increase in muscle mass observed in preclinical models, human clinical trials have shown more modest gains in motor function. This discrepancy likely reflects the chronic denervation and motor neuron loss experienced by many treated patients, which limits the functional translation of muscle hypertrophy. Furthermore, differences in the timing of interventions, neuromuscular plasticity, and species-specific regenerative capacity may contribute to the reduced efficacy seen in humans. These findings highlight the importance of early, combinatorial treatment strategies and the necessity for outcome measures that are sensitive enough to detect subtle improvements in muscle performance.

Emerging data from de Albuquerque et al. and Mackels et al. suggest that circulating myostatin levels may correlate with disease severity and functional impairment, positioning myostatin as a potential biomarker for treatment monitoring and patient stratification [24] (Table 3). While serum myostatin levels offer a feasible and informative proxy of systemic MSTN activity, they may not fully reflect tissue-specific expression patterns, especially in denervated or selectively atrophic muscle groups. The reduced serum concentrations observed in SMA patients likely represent a global consequence of muscle mass loss, rather than localized MSTN production. Therefore, combining serum biomarkers with muscle imaging or regional assessments may enhance our understanding of MSTN-related pathology in SMA. Though preliminary, these insights support a shift toward more personalized medicine approaches in SMA, where treatment strategies could be tailored based on individual molecular or functional profiles [31]. Future Phase 3 clinical trials should aim to combine broad inclusion criteria—to maintain external validity—with predefined stratification based on SMN2 copy number and baseline motor function. Such a design would enable the evaluation of treatment efficacy across the diverse clinical spectrum of SMA, while also facilitating subgroup analyses to identify those most likely to benefit from myostatin inhibition or other adjunctive approaches.

The timing of intervention also appears to be a crucial determinant of treatment efficacy. Several preclinical studies emphasize that earlier administration of myostatin inhibitors—particularly during windows of heightened neuromuscular plasticity—yields more pronounced improvements in NMJ architecture and motor outcomes [27]. However, clinical studies have largely focused on older pediatric and adolescent cohorts, leaving open the question of whether earlier intervention could elicit greater or more durable benefits. Despite the therapeutic promise, several knowledge gaps remain.

First, while 36-month data for Apitegromab are encouraging, longer-term outcomes beyond three years and the durability of motor gains after treatment discontinuation are not yet fully understood. The possibility of compensatory biological mechanisms or tachyphylaxis over extended periods also remains unexplored [26].

Second, the majority of preclinical models represent severe SMA phenotypes. It remains unclear whether these findings extrapolate effectively to milder or atypical SMA presentations or whether efficacy differs significantly by SMA subtype. Additionally, several animal studies did not combine myostatin inhibition with concurrent SMN-targeted therapies, limiting their immediate applicability to clinical settings, where combination regimens are increasingly standard [10].

Third, human studies have yet to systematically evaluate histological and structural endpoints of muscle improvement. While animal models demonstrate increased muscle fiber cross-sectional area and improved NMJ integrity, comparable human data are scarce. Future trials should incorporate muscle biopsies, advanced imaging techniques (e.g., MRI-based muscle volumetry), and electrophysiological assessments to yield more objective correlates of clinical efficacy [36].

Lastly, the potential for additive or synergistic effects with other adjunctive approaches—such as physical therapy, nutritional interventions, or agents addressing inflammation, fibrosis, or mitochondrial dysfunction—warrants systematic investigation. Given the complexity of SMA pathology, a multimodal strategy may be necessary to achieve meaningful and lasting functional recovery in patients with long-standing motor deficits [32].

Although current evidence from preclinical and early-phase clinical studies suggests that myostatin inhibition does not worsen motor neuron loss, the long-term effects of sustained hypertrophic signaling in chronically denervated muscle remain uncertain. Further longitudinal studies are necessary to evaluate whether compensatory muscle growth could impose additional strain on vulnerable neuromuscular units in advanced SMA.

## 4. Conclusions

Taken together, this review supports the clinical and biological plausibility of myostatin inhibition as an adjunctive treatment strategy in SMA. By directly targeting muscle atrophy, myostatin inhibitors complement SMN-enhancing therapies and may help address residual weakness, particularly in patients with established disease. While early clinical results are encouraging, future research must clarify optimal timing, patient selection criteria, long-term efficacy, and the role of combination strategies. Integrating molecular biomarkers and functional imaging may also enhance trial design and therapeutic precision. As SMA care evolves, myostatin inhibition offers a promising new axis for intervention, grounded in strong preclinical science and supported by emerging clinical data.

## 5. Future Directions

Future research should focus on long-term, multicenter clinical trials to assess the sustained effects of myostatin inhibition on motor function and overall quality of life in individuals with SMA. Improved patient stratification—based on SMA subtype, SMN2 copy number, and baseline muscle mass—may enhance therapeutic precision and predictability of outcomes. Incorporating molecular biomarkers and advanced imaging techniques, such as muscle MRI and electromyography, will be essential for establishing robust and objective endpoints. Additionally, the potential for synergistic benefits from combining myostatin inhibitors with adjunctive therapies, such as physiotherapy and nutritional interventions, should be systematically explored.

Moreover, future studies should explore the long-term biological effects of sustained myostatin inhibition, especially in chronically denervated muscle. This involves assessing muscle fibrosis, mitochondrial function, and markers of metabolic strain, which may reveal subtle adverse effects or adaptive responses to prolonged hypertrophic signaling. Integrating advanced imaging techniques, histological analyses, and molecular profiling into long-term clinical cohorts could better define the therapeutic window and long-term safety profile of myostatin-targeted treatments in SMA.

Given the encouraging early evidence, myostatin modulation represents a promising avenue to complement SMN-targeted treatments and expand the therapeutic landscape for SMA.

## Figures and Tables

**Figure 1 ijms-26-05858-f001:**
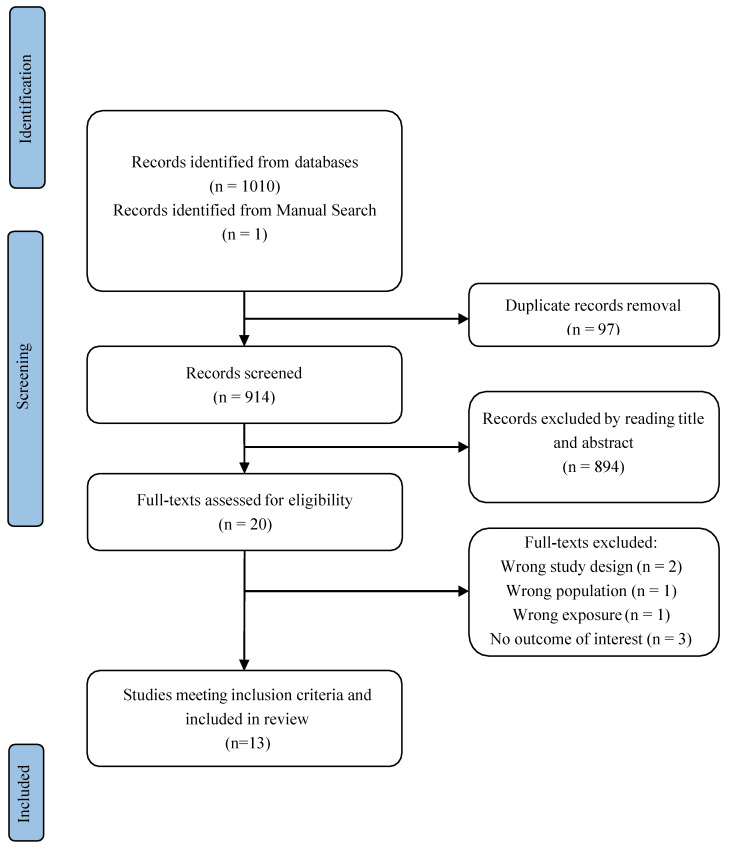
Flow-chart describing the study selection process.

**Figure 2 ijms-26-05858-f002:**
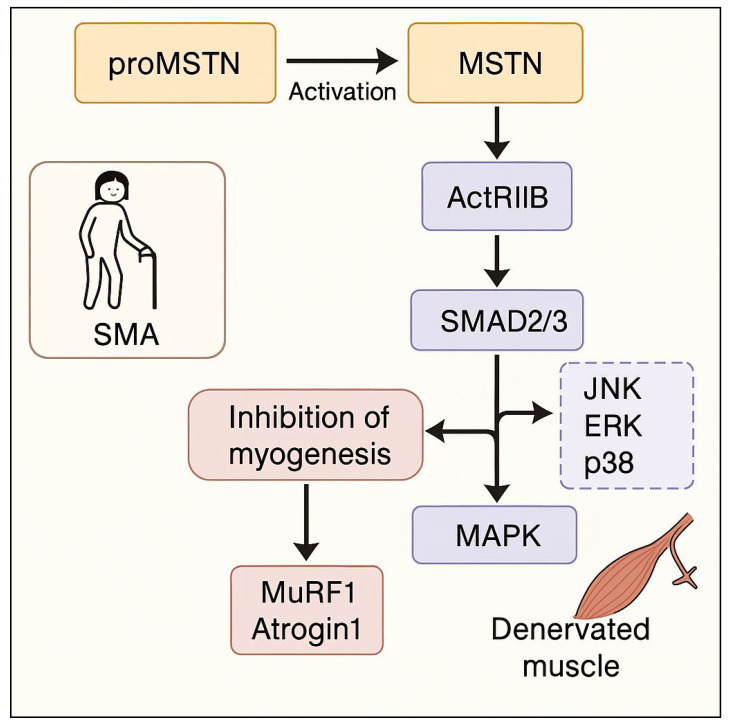
Myostatin-mediated muscle atrophy in SMA. SMA—Spinal Muscular Atrophy; proMSTN—Inactive precursor of Myostatin; MSTN—Myostatin (Growth Differentiation Factor 8, GDF-8); ActRIIB—Activin Receptor Type IIB; SMAD2/3—SMAD Family Members 2 and 3 (transcription factors activated by TGF-β signaling); MAPK—Mitogen-Activated Protein Kinase signaling pathway; JNK—c-Jun N-terminal Kinase; ERK—Extracellular Signal-Regulated Kinase; p38—p38 Mitogen-Activated Protein Kinase; MuRF1—Muscle RING-Finger Protein-1 (muscle-specific E3 ubiquitin ligase involved in atrophy); Atrogin1—Also known as FBXO32, a muscle-specific ubiquitin ligase involved in muscle protein degradation.

**Table 1 ijms-26-05858-t001:** Summary of included preclinical and clinical studies targeting the myostatin pathway in SMA.

Author(s), Year	Country	Study Design	Population Type	SMA Type
Barrett et al., 2021 [26]	USA	Phase 1, randomized, double-blind, placebo-controlled, sequential cohort, two-part, single ascending dose (SAD) and multiple ascending dose (MAD) study.	Healthy adult subjects.	No SMA patients were involved in this study. It aimed to gather safety data to support future trials in SMA patients.
Crawford et al., 2024 [21]	USA	Phase 2 TOPAZ clinical trial.	Human cases.	Types 2 and 3 SMA.
de Albuquerque et al., 2024 [27]	Brazil	Clinical study (cross-sectional case-control followed by 12-month cohort) alongside a bioinformatic study.	Clinical study involved 27 SMA patients and 27 healthy controls. Bioinformatic study involved data from mice, and two studies with humans.	SMA cases were 5q SMA, types 1, 2, and 3.
Feng et al., 2016 [28]	USA	Experimental study.	A7 mice were treated suboptimally with SMN-C3, then subjected to follistatin treatment, in an effort to rescue phenotypes.	This pharmacologically induced model displays features reminiscent of human SMA but presents a unique scenario, enabling a broader examination of what happens. The features of this model indicate a less severe phenotype that shows the pathological responses of human 2 and 3 types.
Liu et al., 2016 [29]	USA	Experimental animal study.	Male C/C SMA model mice.	Type 3 SMA.
Long et al., 2019 [30]	USA	Experimental study.	Used A7 (SMNA7) mouse models.	NA
Mackels et al., 2024 [31]	Belgium and the United Kingdom	Retrospective study.	SMA patients.	Thirteen patients with SMA 1, six patients with SMA 2, and six patients with SMA 3.
Piemonte et al., 2025 [32]	Italy (Multi-center study)	Prospective, multi-center study incorporating cross-sectional and longitudinal components.	Both symptomatic SMA patients receiving DMTs and presymptomatic patients identified through neonatal screening.	Types 2, 2, 3, and presymptomatic.
Rindt et al., 2012 [33]	USA	Experimental model construction and analysis.	Smn−/−; Mstn+/+ Smn−/−; Mstn−/− (or Smn-KO; Mstn-WT) and (Smn- KO; Mstn-KO).	A7 SMA model by genetically invaliding MSTN in conjunction.
Rose Jr et al., 2009 [34]	USA	Experimental study.	hSMN2+/+; hSMN∆7+/+; Smn−/− mice.	NA
Servais et al., 2024 [24]	Belgium, UK, USA	This manuscript reviews the role of myostatin in muscle, explores the preclinical and clinical development of taldefgrobep and introduces the phase 3 RESILIENT trial of taldefgrobep in SMA.	Patients with SMA.	NA
Welsh et al., 2021 [35]	USA	Preclinical toxicology and toxicokinetic (TK) studies.	Cynomolgus monkeys (adult); Sprague Dawley rats (adult and juvenile).	NA
Zhou et al., 2020 [36]	London UK	Experimental animal study, with longitudinal assessment of treated data.	47 mouse models which were constructed from genetic variants to mimic some traits in human SMN.	Severe SMA.

Legend: SMA—Spinal Muscular Atrophy; SMN—Survival Motor Neuron; A7/C/C/SMNA7—Experimental mouse models of SMA; NA—Not Available; DMT—Disease-Modifying Therapy; TOPAZ—Phase 2 clinical trial of Apitegromab; RESILIENT—Phase 3 clinical trial of Taldefgrobep alfa.

**Table 2 ijms-26-05858-t002:** Key outcomes of myostatin inhibition in preclinical and clinical SMA studies.

Author(s), Year	SMA Type	Treatment Route and Dosage	Key Molecular Findings	Functional Outcomes	Histological Outcomes
Barrett et al., 2021 [26]	NA	Intravenous (IV) infusion Part A (SAD): Single ascending doses of apitegromab: 1, 3, 10, 20, 30 mg/kg or placebo. Part B (MAD): Multiple ascending doses of apitegromab: 10, 20, 30 mg/kg or placebo every 2 weeks for a total of three doses.	Dose-dependent and sustained increases in serum latent myostatin were seen.	Apitegromab was safe and well-tolerated. No clinically meaningful changes in baseline vital signs, electrocardiograms, or clinical laboratory parameters and no anti-drug antibody formation.	NA
Crawford et al., 2024 [21]	Types 2 and 3 SMA.	Apitegrob was injected by intravenous treatment every 4 weeks for 12 months.	The elevated target measure also demonstrated quick reached in a dose	Improved effect.	NA
de Albuquerque et al., 2024 [27]	SMA cases were 5q SMA, types 1, 2, and 3.	Data from treated and un-treated individuals analyzed, although treatment with Nusinersen occurred.	Bioinformatic Study: Skeletal muscle gene expression of Mstn decreased and of Fst increased. Clinical Studies: Serum myostatin levels show promise as a novel biomarker for evaluating the severity and progression of spinal muscular atrophy.	A correlation of negative association with clinical severity (CHOP INTEND HFMSE and RULM) was found.	NA
Feng et al., 2016 [28]	This pharmacologically induced model displays features reminiscent of human SMA but presents a unique scenario, enabling a broader examination of what happens. The features of this model indicate a less severe phenotype that shows the pathological responses of human 2 and 3 types.	SMN-C3: Used to initially generate symptomatic adult A7 mice—a suboptimal dose was used to allow survival to adulthood. Regimens reported a greater rescue effect with the higher concentrations during the SMN treatment. After initial findings SMN was stabilized and Recombinant AAV1-follistatin (FS344) then was administered with a dose with PND4 at 5 × 10^11^ Viral particles.	After SMN modification, levels of functional full-length SMN2 were elevated and the treatment was sustained.	Our findings illustrate that this combination improves mean lifespan of transgenic SMA through some sort of rescue mechanism.	Improved or restored synapse and NMJ function as well as an increase in myofiber cross-sectional area.
Liu et al., 2016 [29]	Type 3 SMA.	The 47 mice “were treated intraperitoneally with ~10^12^ genome copies” either via extra-cellular domain protein or w/protease.	Elevated levels of SMAD3; increased CDKN1A.	A7 increased “all limb muscle that is the SMAC or 55”.	NA
Long et al., 2019 [30]	NA	SMN2 modifier SMN-C1 added at PND1 w/IP. Dosage depends on whether it was low or high. Once the mice reached the high-dose concentration, an intermuscular injection of murSRK-015P was given. This concentration and route of action was repeated for 4 weeks.	*After SRK-015 Treatment* vGLUT1 synapses levels as baseline non-SMA Control, *levels of “myostatin are directly linked to activity”*.	When administered to a low-high SMN model, “that the Low-High treatment results in an increase of Synaptic inputs” and increased VHC function as “evidenced by positive trophic factors”.	This report suggests that multiple approaches towards function could be used or targeted to enhance SMN with specific functions in an independent and effective manner.
Mackels et al., 2024 [31]	Thirteen patients with SMA 1, six patients with SMA 2, and six patients with SMA 3.	Intrathecal injection. Doses depended on individual, as research focuses on how levels are altered after specific doses. Data were collected prior to treatment and after 2, 6, 10, 18, and 30 months of treatment.	This found an inverse relation (or that there was decreased volume in key chemicals, as well as their expression).	Lower myostatin levels (meaning more SMN effectiveness) with 32-item Motor Function Measure (MFM32) (*n* = 12, rho = 0.83, 95% CI [0.31; 0.99], *p* < 0.001).	NA
Piemonte et al., 2025 [32]	Types 1, 2, 3, and presymptomatic.	Not specified, but mentioned available DMTs (Nusinersen, Risdiplam, Onasemnogene abeparvovec).	Baseline myostatin levels significantly lower in SMA patients compared to controls. Significant difference in myostatin levels according to functional status in the whole cohort and also across all SMA types. Changes in myostatin in pre-symptomatic before and after treatment.	Significant difference between HFMSE and myostatin levels. Significant difference between CHOP INTEND and myostatin levels.	NA
Rindt et al., 2012 [33]	A7 SMA model by genetically invaliding MSTN in conjunction.	NA	NA	NA	These can be used to identify areas where muscle damage could be detected, and as stated earlier there was little change noted with the process.
Rose Jr et al., 2009 [34]	NA	Recombinant human follistatin administered via intraperitoneal injection.	SMN protein levels in spinal cord and muscle were unchanged in follistatin-treated SMA mice compared with vehicle-treated controls. This indicates an SMN-independent mechanism of action. Plastin-3 levels in spinal cord were similar in follistatin-treated, PBS-treated, and wild-type mice.	Treated animals had improved motor function compared to controls by testing righting reflex, and an increase in mean lifespan by ~30% which equated to 4.6 days.	Ventral horn cell (VHC) number: increased number of VHCs as a result of administration; VHC sections were slightly greater that those of untreated.
Servais et al., 2024 [24]	NA	Taldefgrobep	This study demonstrates little effects of the compound itself, and highlights the complex nature of patients, SMN variants, and function for all treated participants	NA	NA
Welsh et al., 2021 [35]	NA	IV bolus, once weekly. Cynomolgus monkeys: 0, 10, 30, 100 mg/kg; SD rats (adult and juvenile): 0, 30, 100, 300 mg/kg	Confirmed target engagement by measuring serum latent myostatin levels; baseline levels of latent myostatin ranged from approximately 20 to 100 ng/mL. Apitegromab exposure led to higher levels of latent myostatin across multiple timepoints in a dose-dependent manner.	Apitegromab-related increase in muscle weights, ranging to a mean of 3–32% over controls.	Minimal to slight hypertrophy of muscle fibers (limited to right/left biceps brachii), consistent with increased muscle weights.
Zhou et al., 2020 [36]	Severe SMA.	PM025 which acts as an SMN and AAV-MPRO, with subcutaneous injections where both were set between 40 g/g and around a baseline expression.	We can look at full-length SMN transcript, what effects those had in expression and at what rate it occurred. Results from data where “Mstn mRNA level was 80% in skeletal”.	After tests concerning motor skill and function, these areas showed “two-fold increase in time for righting-reflex-skill” also improved with “38% growth of muscle mass”.	After several dissections and “fiber areas” for muscles and neuronal structure results came in with: “a 50% increase” to the neuronal structure’s function,

Legend: IV—Intravenous; IP—Intraperitoneal; PND—Postnatal Day; SAD/MAD—Single Ascending Dose/Multiple Ascending Dose; HFMSE—Hammersmith Functional Motor Scale Expanded; CHOP INTEND—Children’s Hospital of Philadelphia Infant Test of Neuromuscular Disorders; RULM—Revised Upper Limb Module; MFM32—Motor Function Measure, 32 items; MRI—Magnetic Resonance Imaging; TK—Toxicokinetic; AAV—Adeno-Associated Virus; SMAD3—Mothers Against Decapentaplegic Homolog 3; CDKN1A—Cyclin-Dependent Kinase Inhibitor 1A; Fst—Follistatin; Mstn—Myostatin; NA-Not applicable.

**Table 3 ijms-26-05858-t003:** Comparative summary of key quantitative outcomes from preclinical and clinical studies on myostatin inhibition in SMA.

Outcome	Preclinical Studies	Clinical Studies
% Increases in muscle mass	Up to +38% muscle mass increase in mouse models (Zhou et al., 2020) [36]; +3–32% in primate and rodent models (Welsh et al., 2021) [35].	Muscle mass increase not directly quantified; implied via improved strength and motor scores (TOPAZ study).
Motor function improvement	Improved righting reflex and lifespan (+4.6 days) in SMA mice (Rose et al., 2009) [34]; enhanced synaptic inputs (Long et al., 2019) [30].	HFMSE improvement of +3.6 points at 12 months, sustained at 36 months (Crawford et al., 2024 [21]; TOPAZ trial).
NMJ Architecture/Function	Enhanced NMJ integrity and vGLUT1 synapse density (Long et al., 2019) [30]; increased cross-sectional area of myofibers (Feng et al., 2016 [28]).	NMJ structure not assessed directly; clinical improvements observed, but structural endpoints not reported.
Histological/Structural outcomes	Muscle fiber hypertrophy and preserved motor neurons (Rose et al., 2009 [34]; Feng et al., 2016 [28]).	Structural outcomes not evaluated; no histological data available.
Serum myostatin modulation	Reduction in Mstn mRNA and increase in Fst in muscle (de Albuquerque et al., 2024 [27]).	Dose-dependent increase in serum latent myostatin after Apitegromab (Barrett et al., 2021 [26]; Mackels et al., 2024 [31]).
Duration and persistence of effect	Short-term survival benefit and structural rescue in animal models (Rose et al., 2009 [34]; Zhou et al., 2020 [36]).	Motor gains sustained over 36 months in SMA types 2 and 3 (Crawford et al., 2024 [21]).

Legend: HFMSE—Hammersmith Functional Motor Scale Expanded; NMJ—Neuromuscular Junction; vGLUT1—Vesicular Glutamate Transporter 1; Fst—Follistatin; Mstn—Myostatin.

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
