# Peer review of "Myostatin Modulation in Spinal Muscular Atrophy: A Systematic Review of Preclinical and Clinical Evidence"

_ijms, 2025, doi:10.3390/ijms26125858_

Round 1

Reviewer 1 Report

Comments and Suggestions for Authors

The authors present a survey on the effects of myostatin inhibition in the treatment of spinal muscular atrophy (SMA). The systematic literature review screened all original research articles on SMA and myostatin (growth differentiation factor 8). The final selection includes 13 publications published in the years 2009 to 2025. The review synthezises findings from preclinical and clinical studies investigating myostatin modulation in SMA. Preclinical investigation, employing the C/C or A7 mouse models of SMA, constantly show improved muscle mass and function through myostatin pathway inhibition. The gain in muscle strength in clinical trials is less impressing, but the studies have shown favourable safety profiles. 

As a side effect it has been shown that  myostatin levels in circulating blood correlate with disease severity and functional impairment, making myostatin a potential biomarker for treatment monitoring and patient selection. 

The article is well written, the selection of literature is well explained, the results and the discussion are clear. 

The topic is relevant and urgent. The article should be published to stimulate further studies and clinical trials.

Reviewer 2 Report

Comments and Suggestions for Authors

This review article provides a systematic and in-depth analysis of the potential of myostatin inhibition as an adjunctive strategy for treating Spinal Muscular Atrophy (SMA). The scholarly value of the work is high, as it synthesizes both preclinical and clinical evidence and highlights the therapeutic gap left by SMN-targeted therapies alone. However, several aspects require revision to further strengthen the manuscript.

 Introduction

  1. What specific pathophysiological evidence led the author to prioritize myostatin inhibition over other muscle-enhancing pathways, such as IGF-1/AKT signaling, in the context of SMA?

  2. How does the proposed dual targeting of SMN and myostatin address the distinct mechanisms of motor neuron loss and muscle atrophy in SMA?

Results

  1. On what basis were the SMA type II/III subgroups selected for the Phase II Apitegromab trial, given the heterogeneity in SMN2 copy numbers among patients?

  2. Why did the substantial increase in muscle mass observed in SMA mouse models not translate into proportional improvements in motor function during human clinical trials?

  3. How do serum myostatin levels, as reported in recent studies, account for tissue-specific expression patterns of MSTN in SMA patients?

  4. What is the rationale for selecting anti-proMyostatin antibodies, such as Apitegromab, over latent myostatin-targeting agents like GYM329 in ongoing clinical trials?

Conclusion

  1. Should future Phase III clinical trials prioritize broader patient inclusion criteria or stricter stratification based on SMN2 copy number and baseline motor function?

  2. Could myostatin inhibition potentially exacerbate compensatory muscle hypertrophy mechanisms, thereby accelerating denervation in advanced cases of SMA?

Round 2

Reviewer 2 Report

Comments and Suggestions for Authors

1. Authors mention the uncertainty regarding long-term myostatin inhibition effects in chronically denervated muscle. Could future directions include evaluating fibrosis, mitochondrial function, or metabolic strain in long-term cohorts to better define therapeutic windows and safety?

2. While authors' explanation of the translational gap is reasonable, would it be possible to include a comparative summary (table or figure) that contrasts key quantitative outcomes from preclinical vs. clinical studies (e.g., % increase in muscle mass, NMJ changes, motor scores)? This may strengthen the translational insight of the review.

3. While the rationale for preferring Apitegromab is well explained mechanistically, are there any comparative safety or tolerability data from early-phase studies that quantitatively support its selection over latent-targeting agents such as GYM329 or Taldefgrobep?

Round 3

Reviewer 2 Report

Comments and Suggestions for Authors

The revised manuscript shows significant improvement. The authors have adequately responded to the previous feedback and the work now meets the publication standards.